# Molecularly Guided Drug Repurposing for Cholangiocarcinoma: An Integrative Bioinformatic Approach

**DOI:** 10.3390/genes13020271

**Published:** 2022-01-29

**Authors:** Simran Venkatraman, Brinda Balasubramanian, Pisut Pongchaikul, Rutaiwan Tohtong, Somchai Chutipongtanate

**Affiliations:** 1Graduate Program in Molecular Medicine, Faculty of Science Joint Program Faculty of Medicine Ramathibodi Hospital, Faculty of Medicine Siriraj Hospital, Faculty of Dentistry, Faculty of Tropical Medicine, Mahidol University, Bangkok 10400, Thailand; simran.ven@student.mahidol.ac.th (S.V.); balasubramanian.bri@student.mahidol.ac.th (B.B.); 2Chakri Naruebodindra Medical Institute, Faculty of Medicine Ramathibodi Hospital, Mahidol University, Samut Prakan 10540, Thailand; pisut.pon@mahidol.edu; 3Institute of Infection, Veterinary and Ecological Sciences, University of Liverpool, Liverpool L69 7ZX, UK; 4Department of Biochemistry, Faculty of Science, Mahidol University, Bangkok 10400, Thailand; 5Pediatric Translational Research Unit, Department of Pediatrics, Faculty of Medicine Ramathibodi Hospital, Mahidol University, Bangkok 10400, Thailand; 6Department of Clinical Epidemiology and Biostatistics, Faculty of Medicine Ramathibodi Hospital, Mahidol University, Bangkok 10400, Thailand

**Keywords:** cholangiocarcinoma, connectivity map, drug–gene network, drug repurposing, immune-oncogenic gene signature, transcriptomics, survival analysis

## Abstract

Background: Cholangiocarcinoma (CCA) has a complex immune microenvironment architecture, thus possessing challenges in its characterization and treatment. This study aimed to repurpose FDA-approved drugs for cholangiocarcinoma by transcriptomic-driven bioinformatic approach. Methods: Cox-proportional univariate regression was applied to 3017 immune-related genes known a priori to identify a list of mortality-associated genes, so-called immune-oncogenic gene signature, in CCA tumor-derived RNA-seq profiles of two independent cohorts. Unsupervised clustering stratified CCA tumors into two groups according to the immune-oncogenic gene signature expression, which then confirmed its clinical relevance by Kaplan–Meier curve. Molecularly guided drug repurposing was performed by an integrative connectivity map-prioritized drug-gene network analysis. Results: The immune-oncogenic gene signature consists of 26 mortality-associated immune-related genes. Patients with high-expression signature had a poorer overall survival (log-rank *p* < 0.001), while gene enrichment analysis revealed cell-cycle checkpoint regulation and inflammatory-immune response signaling pathways affected this high-risk group. The integrative drug-gene network identified eight FDA-approved drugs as promising candidates, including Dasatinib a multi-kinase inhibitor currently investigated for advanced CCA with isocitrate-dehydrogenase mutations. Conclusion: This study proposes the use of the immune-oncogenic gene signature to identify high-risk CCA patients. Future preclinical and clinical studies are required to elucidate the therapeutic efficacy of the molecularly guided drugs as the adjunct therapy, aiming to improve the survival outcome.

## 1. Introduction

Cholangiocarcinoma (CCA) is a highly prevalent biliary malignancy that is notoriously heterogeneous and is usually diagnosed at advanced stages [1]. CCA possesses a complex tumor microenvironment (TME) that can regulate the occurrence and progression of cancer. Cancer-associated fibroblasts, tumor-associated macrophages, myeloid-derived suppressor cells, stromal cells, and cytokines promote an immunosuppressive environment to foster CCA progression. Infiltration of these cells in CCA is associated with poor survival outcomes [2,3]. Due to its multifaceted tumor architecture and heterogeneous nature, current treatment options cannot mitigate CCA progression effectively.

Recent advents in cancer immunology led to the development of novel therapies, including immune checkpoint blockades (ICBs) against PD-1 and CTLA-4, chimeric antigen receptor (CAR) T cells, and recombinant cancer vaccines [4]. These therapies have been successful; however, the response rate varies among patients and cancer types. In CCA, Pembrolizumab exhibited anti-tumor activity in 6–13% of advanced-stage patients [5]. Additionally, the phase 1 trial of epidermal growth factor-specific CAR T cell therapy in unresectable CCA showed that of 17 evaluable patients, one patient achieved complete remission, ten patients had stable disease [6]. Varied responses to immunotherapy are owed to the heterogeneous nature of immunosuppressive microenvironments and the expression of immune-related genes in tumors [7]. Hence, this warrants molecular characterization of tumors in an immune context.

This bioinformatic study aimed to identify immune-related genes that recognized the high-risk CCA patients with a poorer prognosis and repurposing FDA-approved drugs that could be beneficial for the high-risk CCA patients. The entire workflow of this study is shown in Figure 1.

## 2. Materials and Methods

### 2.1. Data Acquisition and Pre-Processing

Transcriptomic profiles of patient-derived CCA tumor tissues were collected from two independent cohorts. For the first cohort, RNA-seq data were obtained from Gene Expression Omnibus (GEO; accessed on 24 June 2021) using the search terms “cholangiocarcinoma” AND “human” AND “RNA-seq”. Results were filtered to select only processed RNA-seq data, resulting in the reads per kilobase million (RPKM) counts matrix from GSE107943, containing 30 CCA samples with 27 matched normal liver tissues. For the second cohort, RNA-seq data of TCGA-CHOL was acquired from cBioportal (www.cbioportal.org; accessed on 24 June 2021), containing 36 CCA tumor samples. Each dataset was treated independently. For the validation dataset, we applied our previous compilation of CCA tumor microarray profile containing 704 tumors from 10 independent cohorts (i.e., GSE132305, GSE22633, GSE26566, GSE32225, GSE32879, GSE35306, GSE57555, GSE66255, GSE76279, and GSE89749) [8]. The immune-oncogenic signature was derived from Thorsson V et al., study [9] containing 3017 genes shown in Appendix A.

### 2.2. Survival Analysis

Survival analysis was performed using the survival, survminer, and survplot packages in R version 4.0.2. Cox-Proportional Univariate Regression model was applied to each gene in the immune-oncogenic signature. Hazard Ratios (HR) and Wald Statistic *p*-value were tabulated and plotted. Clinically relevant genes in the immune-oncogenic signature were filtered using the cut-offs of HR > 1 and *p*-value < 0.05. These candidate genes were fitted to the Cox regression curve. Venn diagrams of common candidates were drawn from http://bioinformatics.psb.ugent.be/webtools/Venn/ (accessed on 25 August 2021). 

### 2.3. Unsupervised Hierarchical Clustering

Unsupervised hierarchical clustering was conducted to explore whether the filtered gene signature expression can stratify CCA patients. Heatmap was drawn using the pheatmap package. Samples were then grouped according to the clusters corresponding to the low- and high-expression of the immune-oncogenic gene signature.

### 2.4. Differential Gene Expression Analysis

To assess the transcriptomic differences between samples with low and high expression of the filtered gene list. Principal Component Analysis and differential gene expression analysis was conducted using the DESeq2 package for the RNA-Seq datasets and limma package for the Microarray datasets in R. *p*-values were determined by Wald statistics and an adjusted *p*-value (*Q*-value) to correct for multiple comparisons testing using the Benjamini–Hochberg method. DEGs were defined as genes with 2× fold-change and adjusted *p*-value < 0.05. 

### 2.5. Pathway Enrichment Analysis

The DEGs were then input into EnrichR [10] to evaluate their significance in biological pathways using the BioPlanet 2019, WikiPathways 2021, and Kyoto Encyclopedia of Genes and Genomes (KEGG) 2021 databases. *p*-values were calculated using Fisher’s exact test, and enrichment ‘Combined’ scores were calculated by combining the *p*-value and z-score. The enrichment results were plotted using the ggplot2 package in R. 

### 2.6. Pharmacogenomic Connectivity Analysis 

The DEGs and their respective log2-fold change and adjusted p-values served as the input for pharmacogenomic connectivity analysis using the integrative LINCS L1000 database portal [11,12] (ilincs.org; accessed on 25 August 2021). Using a query analysis, we input our DEG signature of up- and down-regulated genes and computed the connectivity levels between our signature and that of the chemical perturbagens of the LINCS database using random query analysis. Chemical perturbagens with “unusually” high similarity scores were tabulated [11,12]. The identified chemical perturbagens were considered valid when *p*-value < 0.01.

### 2.7. Drug–Gene Network Analysis

STITCH, the Search Tool for Interactions of Chemicals, is a web-server for identifying interactions between the defined genes and perturbagens from text-mining, experimental evidence, co-expression, gene-fusion, and database annotation sources [13]. The common connected perturbagens and the CCA immune-oncogenic gene signature were input into STITCH-DB (http://stitch.embl.de; accessed on 25 August 2021) selecting Homo sapiens as the organism identifier. Default parameters were maintained in the network visualization. These node and edge data was tabulated and re-plotted using Cytoscape v.3.8. Nodes were re-annotated using the stringApp v.1.7.0 plugin.

## 3. Results

### 3.1. Deriving the Mortality-Associated Immune-Related Genes for CCA

The comprehensive list of immune-oncogenic genes was retrieved from Thorsson V et al. [9]. This list is comprised of 160 immune expression signatures from text-mining sources and gene-set databases. When summarized, the resulting gene signature consisted of 3017 genes (Appendix A). These genes were significantly enriched in T-cell receptor signaling, cytokines and inflammation, pathways involved in autoimmune diseases and immune responses, and the Interleukin-STAT signaling pathway (Appendix A). 

To identify which immune-related genes are associated with patient prognosis, the Cox Proportional Univariate regression model was applied to CCA patients of both cohorts, GSE107943 and TCGA-CHOL, assessing the effect of each gene in the list against the overall survival. The mortality-associated gene was considered valid when its Hazard Ratio (HR) > 1 and *p*-value < 0.05. The resulting gene list comprised of 386 genes in the GSE107943 cohort (Figure 2a and Appendix A) and 103 genes in the TCGA-CHOL cohort (Figure 2b and Appendix A). Here, a Venn diagram showed that 26 genes were consistently recognized between two independent cohorts of CCA patients (Figure 2c and Appendix A). The common 26 genes were significantly enriched in oncogenic pathways, e.g., TP53 network and PI3K/AKT/mTOR signaling (Appendix A), that linked to CCA oncogenesis and chemoresistance [1]. The molecularly derived list of 26 mortality-associated immune-related genes, so-called the immune-oncogenic gene signature, was then applied throughout this study.

The unsupervised hierarchical clustering was performed to stratify CCA samples depending on the expression patterns of the immune-oncogenic gene signature. Accordingly, 18 vs. 12 samples in the GSE107943 (Figure 3a), and 23 vs. 13 samples in the TCGA-CHOL (Figure 3b), were recognized as the low- vs. the high-expression groups, respectively. Next, the Kaplan–Meier curve was plotted to evaluate the overall survival rate of CCA patient subpopulations as stratified by the immune-oncogenic gene signature (total *n* = 66; 41 low- vs. 25 high-expression). Interestingly, patients in the high-expression group were significantly associated with a poorer overall survival (log-rank *p*-value < 0.001) (Figure 3c). These findings suggested that intertumoral heterogeneity of CCA driven by the immune-oncogenic gene signature associated with the patient prognosis. Further analysis of transcriptomic changes between CCA patients with the low-expression (owning a better prognosis) and the high-expression (owning a poorer prognosis) signature could provide insights into mechanistic pathways and therapeutic targets driven by the immune-oncogenic gene signature. 

### 3.2. CCA Immune-Oncogenic Gene Signature Involved in Various Cancer Signaling Pathways

To gain insight into mechanistic pathways underlying the high-risk CCA patient subpopulation (the high-expression group), we performed differential expression analysis between 12 low- vs. 18 high-expression of immune-oncogenic gene signature in the GSE107943 cohort, resulting in 150 differentially expressed genes (DEGs) with 2× fold-changes and adjusted *p*-value < 0.05 (Appendix A). Pathway enrichment analysis showed that DEGs were significantly enriched in cell-cycle checkpoint regulation and cytokine signaling pathways, i.e., interferon alpha/beta, oncostatin M (an IL-6 family member), IL-4, and TGF-β (Appendix A). The cell-cycle checkpoint molecule CyclinD-CDK4 complex regulated the immune checkpoint PD-L1 expression [14], while the anti-PD-L1 therapy potentiated the effect of CDK4/6 inhibitors [15]. Interleukins and type 1 interferon signaling established chronic inflammation to drive immune evasion, cancer progression and metastasis in colorectal, pancreatic, and hepatocellular carcinoma [16,17]. 

To ensure this phenomenon is generalizable in patients with CCA, the differential expression and pathway enrichment analyses were again performed on TCGA-CHOL cohort (13 low- vs. 23 high-expression). As a result, a total of 888 DEGs were identified (Appendix A). Pathway enrichment analysis demonstrated DEGs were significantly enriched in the interleukins and TGF-β signaling pathways. Additionally, EGFR signaling pathway and coagulation and complement signaling were also significantly enriched (Appendix A). These results were in line with our previous findings (Appendix A), implying that the immune-oncogenic gene signature governs immune responses and various oncogenic signaling pathways associated with CCA proliferation and progression (Appendix A). Addressing the molecular immune-oncogenic machinery in the high-risk CCA patients could guide to an effective therapeutic strategy. 

### 3.3. Drug Repurposing for the High-Risk CCA Patients 

To repurpose FDA-approved drugs that may improve therapeutic outcomes in the high-risk CCA patients, we performed pharmacogenomic connectivity analysis [11] between CCA tumor-transcriptional changes driven by the immune-oncogenic gene signature (the low- vs. high-expression groups) with L1000-based cellular transcriptional changes induced by thousands of chemical perturbagens. The DEGs (150 and 888 genes derived from the GSE107943 and TCGA-CHOL cohorts, respectively) with their corresponding fold-changes and p-values were used as the input signature to survey for chemical perturbagen signatures positively correlated to the input. The results showed that 268 and 949 small molecules were significantly matched for the GSE107943 and TCGA-CHOL cohorts, respectively (full details in Appendix A). The top 25 predicted compounds for GSE107943 and TCGA-CHOL, ranked by the connectivity z-score, are shown in Figure 4a,b, respectively. Notably, in the GSE107943, several cyclin-dependent kinase inhibitors (including dinaciclib, alvocidib, and seliciclib) were discovered for their potential effects on CCA transcriptomic reversal. This observation was maintained in the TCGA-CHOL cohort, but additionally several tyrosine-kinase inhibitors (i.e., dacomitinib and pazopanib) were also identified. A Venn diagram of the candidate small molecules from the analyses of GSE107943 and TCGA-CHOL datasets identified 29 mutual entities. These small molecules belong but are not limited to the classes of kinase inhibitors (for dasastinib, selumetinib, and trametinib), cyclin-dependent kinase inhibitors (for AT-7519 and BMS-387032), and epigenetic modulators (for pracinostat) (Figure 4c,d and Appendix A). 

### 3.4. Pharmacogenomic Connectivity and the CCA Immune-Oncogenic Gene Signature

Finally, we determined the potential causal effects between the transcriptomic profile-guided drugs and immune-oncogenic signals driven by the immune-oncogenic gene signature. Accordingly, the drug-gene interactome of 29 predicted drugs (as shown in Appendix A) and 26 mortality-associated immune-related genes in CCA patients (Figure 2c) was generated by using the STITCH database [13] with the Cytoscape v.3.8 and stringApps v.1.7.0 plugin. As a result, the predicted drugs established several meaningful interactions to the members of the immune-oncogenic gene signature (Figure 5 and Table 1). 

Most interactions converged at dasatinib (a multi-kinase inhibitor), which directly targets ABL1, FRK, FYN, LCK, and SRC, five gene members of the immune-oncogenic gene signature. Dasatinib interacted with a cluster of tyrosine kinase inhibitors, including canertinib (EGFR inhibitor), and selumetinib (MEK1/2 inhibitors). These results were consistent with the enriched EGFR1 and FGF signaling pathways (Appendix A) and involved with the MAPK signaling cascade. Bexarotene interacts with MDM2 by promoting its p53-mediated gene expression [18]. Bexarotene has been implicated cancer immunity and has been approved for the treatment of skin manifestations of cutaneous T-cell lymphoma. Moreover, NVP-TAE226 inhibits the AURKA/AKT/FAK signaling pathway by inhibiting FAK, thereby waning cell invasion and migration in head-and-neck squamous cell carcinoma [19]. Additionally, PF-573228, another FAK inhibitor, interacts with SRC by impeding the formation of the SRC/FAK complex and integrin activation through the inhibition of FAK, resulting in reduced cell proliferation in thyroid cancer [20]. These results provide insight into the causal effect of predicted drug action through cell-cycle regulation and FAK signaling pathways. Cyclin-dependent kinase (CDK) inhibitors were not directly connected to the immune-oncogenic gene signature; however, TP53 and AURKA have shown evidence in regulating CDK activity [21]. Additionally, several interactions were established around the members of the immune-oncogenic gene signature and TP53, BRCA1, and AURKA, which involved in regulating spindle assembly and the cell-cycle [21], and TGF-β signaling [22], thus supporting the potential of CDK inhibitors against the immune-oncogenic related pathways behind the high-risk CCA phenotype (Figure 4c and Figure 5c). Taken together, the molecularly driven pharmacogenomic connectivity-mapping with the integrative drug-gene network analysis successfully delivered 8 FDA-approved drugs and 11 investigational drugs in multiple phases of clinical trials (Table 1) as promising candidates for adjunct treatment to improve therapeutic outcomes in high-risk CCA patients.

### 3.5. Validation Analysis Using a Pooled CCA Cohort

To extend this generalizability across platforms, we validated the use of the signature to stratify 704 CCA tumors with microarray profiles compiled from 10 independent cohorts and previously reported by our group [8]. We found that patient samples stratified themselves into groups of low-, intermediate-, and high- expressing samples (268 low, 182 intermediate, and 254 high expressing samples) (Appendix A). When differential expression and pathway enrichment analysis was conducted, we found 4765 genes differentially expressed with 2× fold-changes and adjusted *p*-value < 0.05 (Appendix A). The pathway enrichment of these genes revealed interferon related signaling. Additionally, DNA damage, gene expression, and protein metabolism pathways were also significantly enriched. These pathways are consistent with our findings in the GSE107943 and TCGA-CHOL datasets (Appendix A). Lastly, the connected perturbagens identified several kinase inhibitors such as: Imatinib, NVP-AEW541, and BMS-777607. Moreover, in indirectly targeting elements of the CCA immune-oncogenic gene signature, SJ 172550, an MDM4 was also significantly enriched. An epigenetic modulator was also identified, including PCI-34051 (Appendix A). Interestingly, the perturbagen identified commonly among the three analyses was WH-4-025, a dual LCK/SRC tyrosine kinase inhibitor (Appendix A). These findings provide significance to the immune-oncogenic gene signature, in that it was capable of stratifying patients, and identifying positively correlated connected perturbagens. 

## 4. Discussion

While elements of the tumor immune microenvironment may be shared across various tumor types, the composition and mechanism of these components are unique to each tumor lineage. This presents a challenge in characterizing tumors based on infiltration levels or transcriptomic signatures defined globally to all tumor types. To address this issue, the present investigation derived a clinically relevant onco-immune signature to characterize and stratify CCA patients and utilize this CCA sample stratification strategy to identify small molecules that can be repositioned to potentially combat CCA and its complex immune architecture. 

Here, we formulated the immune-oncogenic gene signature containing 26 mortality-associated immune-related genes that stratified CCA samples according to the molecular signature expression patterns. The similar strategy had been employed with successes to identify tumor-lineage-specific immune signatures in lung adenocarcinoma [23] and breast cancer [24]. However, the present study was the first effort, to the best of our knowledge, to focus on drug discovery and repurposing by the immune-oncogenic gene signature-driven pharmacogenomic connectivity mapping and comprehensive network analysis. 

The immune-oncogenic gene set provided by Thorsson V et al. [9], comprised of 3017 genes, was seen to be enriched in various aspects of immune responses. These include cytokines and inflammatory response, T-cell receptor activation, NO2 dependent IL-12 pathway for NK cells, macrophage markers, and bystander B-cell activation (Appendix A). These inflammatory signaling pathways have been associated with tumor initiation, angiogenesis, and metastasis of cancers, by modulating TME [25]. Moreover, in the process of immune evasion, several immune cell types, such as macrophages, NK cells, T cells, and B cells, work in concert to create an immunosuppressive TME [26]. Nonetheless, one should be aware that not all elements of the immune-oncogenic gene set are clinically relevant for CCA. This study then sorted out a subset of this immune-oncogenic gene signature as clinically relevant genes (*n* = 26 genes) commonly presented in two independent cohorts and associated with the poorer prognosis when highly expressed in CCA tumors (Figure 2). The gene members of this immune-oncogenic gene signature still retained the oncogenic pathways through mitotic spindle formation, PI3K-Akt-mTOR signaling, and Aurora B signaling (Appendix A). These pathways are known to implicate in the regulation of anti-tumor immunological surveillance. For instance, PI3K-Akt-mTOR pathway is reported to be essential in regulating the secretion of immunosuppressive cytokines such as TGF-β and IL-10 [27]. These findings suggested that 26 gene members of the immune-oncogenic gene signature might play crucial roles in CCA oncogenic processes, regulating the cell cycle or modulating immunosuppressive TME.

Pathway enrichment analysis of CCA transcriptional changes between the low- vs. high-expression signature groups revealed the critical involvements of cell-cycle regulation, mitotic spindle formation, complement and coagulation signaling cascade, and cytokine signaling pathways (i.e., interferons and IL-1) in the immune-oncogenic processes of the high-risk CCA phenotype (Appendix A). These pathways have been previously observed for their roles in various cancer types. Regulation of the cell cycle has implications in anti-tumor immunity [28,29] contributed by AURKA, a member in the immune-oncogenic gene signature. AURKA regulates the cell-cycle through the p53 and NF-kB pathway [30]. In melanoma, the combination of AURKA inhibitors and MDM2 inhibitors synergistically promoted anti-tumor immune cell infiltration in immunocompetent mice [31]. Activation of the complement system induces the accumulation and differentiation of various tumor-associated neutrophils. These neutrophils release proteinases which promote tumor growth, as witnessed in lung cancer [32]. CCA tumor cells secrete several cytokines to establish the immunosuppressive TME [2]. To modulate the tumor microenvironment, several cytokine-targeted therapies have been coupled with the standard-of-care chemotherapy, such as the use of Interferon-alpha with G-colony stimulating factor, fluorouracil, and hydroxyurea (Clinicaltrials.gov identifier NCT00019474). Interferon-alpha 2 is a cytokine that promotes antigen presentation via upregulation of MHC-I/II, leading to increased infiltrations of dendritic cells and T cells into melanoma tumors [33]. Similarly, clinical trials have been conducted in CCA patients to evaluate the efficacy of interferon-alpha 2 with an immune checkpoint blockade pembrolizumab (Clinicaltrials.gov identifier NCT02982720), or in combination with standard-of-care chemotherapy (Clinicaltrials.gov identifier NCT00019474). This line of evidence supports that the immune-oncogenic processes in CCA, as driven by the immune-oncogenic gene signature, deserve attention as promising therapeutic targets.

In this direction, drug discovery and repurposing by the pharmacogenomic approach, coupled with integrative network analysis, predicted 29 candidate small molecules which might intervene the immune-oncogenic signals in CCA tumors with the high-expression signature (Figure 5 and Table 1). These candidate perturbagens are primarily enriched in FDA-approved tyrosine kinase inhibitors. Among these, dasatinib was recognized as the most promising candidate. Dasatinib directly interacts with 5 out of 26 genes in the immune-oncogenic gene signature (i.e., ABL1, FYN, LCK, SRC, and FRK), and also links to TP53 which is involved in regulating cell-cycle [21] and TGF-β signaling [22]. Multiple targets of dasatinib action, particularly SRC [34] and ABL [35], are crucial in CCA progression. In fact, several tyrosine kinase inhibitors have potential to modulate inflammatory responses within the TME which can affect cancer immunotherapy [36]. While dasatinib is recognized as the suitable candidate due to its targeting multiple kinases involved in immune-oncogenic pathways, it shows promise in modulating the tumor immune microenvironment as well. Dasatinib treatment increased Th1 and CD8+T cell levels in patients with chronic myeloid leukemia who had better therapeutic responses [37]. Currently, dasatinib is being investigated for isocitrate dehydrogenase (IDH)-mutant advanced intrahepatic cholangiocarcinoma (NCT02428855). Considering this finding, and the potential dasatinib holds in treating CCA, future explorations can unravel the anti-tumoral immunogenic effects of dasatinib in CCA patients who express high levels of the immune-oncogenic gene signature. 

Another drug class in the network that deserved our attention is CDK inhibitors, including AT-7519 and BMS-387032 (Figure 5). AT-7519 is the second generation multi-CDK inhibitor that targets CDK1/2/4/6/9 and has proven effective in inhibiting hematologic malignancies [38], while BMS-387032 is a selective CDK2 inhibitor that has shown promise in treating acute myeloid leukemia in vitro [39]. It was known that CCA cell lines were ubiquitously reliant on CDK4/6 activity for cell proliferation, and therefore were responsive to CDK4/6 inhibition [40]. Additionally, recent reports revealed the use of CDK inhibitors in triggering immunogenic responses in cancers by destabilizing PD-L1 [14] and enhancing anti-tumor immunity through the production of type III interferon and the suppression of regulatory T cells [15]. Nonetheless, such evidence demonstrating AT-7519 and BMS-387032 in regulating anti-cancer immune functions in the context of CCA is lacking. Our findings support further investigations, both to explore the efficacy of AT-7519 and BMS-387032 in CCA, and unravel its mechanism in an immune context. 

This study had limitations. First, the CCA immune-oncogenic gene signature was derived from a modest sample size, and has not been explored for its performance as a multivariate signature. However, this constraint encouraged us to validate all findings in two independent RNA-Seq gene expression cohorts (GSE107943 and TCGA-CHOL), and ten independent microarray cohorts, which collectively showed expression of this gene signature in varying profiles. Nevertheless, future investigations may be directed towards performing multivariate and LASSO regression analysis to assess the performance of the immune-oncogenic gene signature as a prognostic marker. Second, this study was limited by the nature of bioinformatic study in which the conclusion had no support from the experimental validation. Nonetheless, different methods applied in this study provided the consistent results and thus serve as a rationale for further investigations of the candidate drugs (Table 1) for the experimental evidence of its efficacy in preclinical models and clinical studies. For instance, WH-4-025, the dual LCK/SRC inhibitor, was identified as the commonly enriched perturbagen across 12 independent CCA cohorts (Appendix A). Hence, the present study offers an opportunity to explore and reposition several candidates to be developed in preclinical and clinical models for the treatment of CCA. Moreover, future investigations may also explore the contexture of the CCA immune-oncogenic signature in the tumor immune microenvironment by comparing the expression levels between the tumor and its microenvironment using spatial or single-cell transcriptomics.

## 5. Conclusions

In conclusion, this study identified the immune-oncogenic gene signature consisting of 26 mortality associated immune-related genes that consistently presented in two independent cohorts. Immune-oncogenic gene signature-driven patient stratification could identify high-risk CCA patients with a poorer prognosis. Drug repurposing by pharmacogenomic connectivity mapping with drug-gene network analysis predicted several FDA-approved kinase inhibitors. Future investigations are warranted to evaluate therapeutic potentials of the predicted drugs as the adjunct to the standard treatment regimens for high-risk CCA patients, aiming to improve the survival outcome.

## Figures and Tables

**Figure 1 genes-13-00271-f001:**
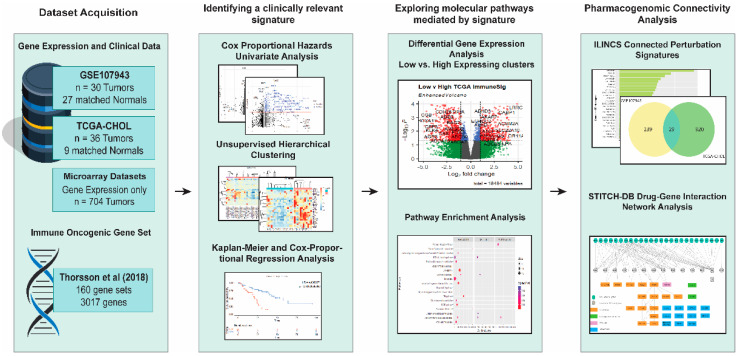
Schematic workflow of the present study. Vectors designed by FreePik (accessed on 30 August 2021).

**Figure 2 genes-13-00271-f002:**
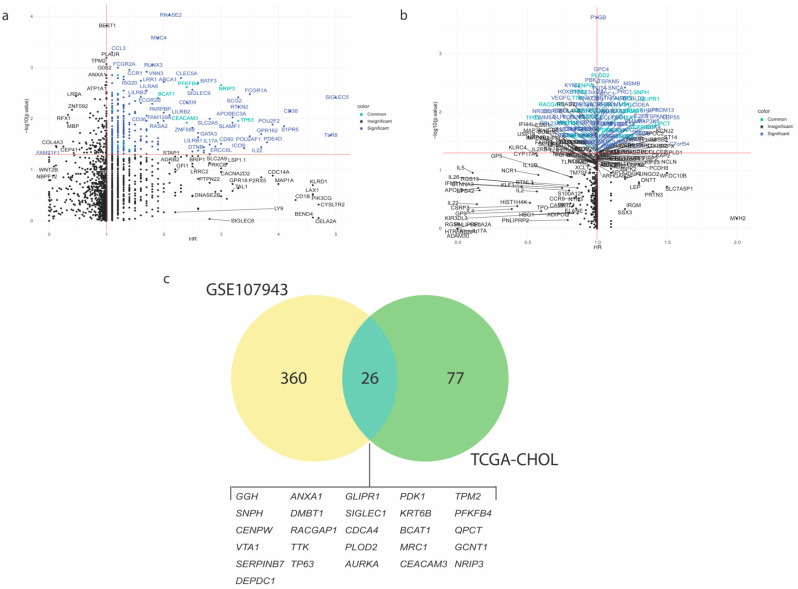
Identification of the mortality-associated immune-related genes for cholangiocarcinoma (CCA). The scatter plots show Cox-Proportional Hazard Ratio (HR; x-axis) and −log10 (*p*-value) (y-axis) of 3017 immune-related genes in CCA tumor transcriptomic profiles from (**a**) GSE107943 (*n* = 30 samples) and (**b**) TCGA-CHOL (*n* = 36 samples) cohorts. The candidate gene was considered valid if HR > 1 and *p*-value < 0.05. (**c**) Venn diagram shows 26 mortality-associated immune-related genes that commonly presented between two cohorts.

**Figure 3 genes-13-00271-f003:**
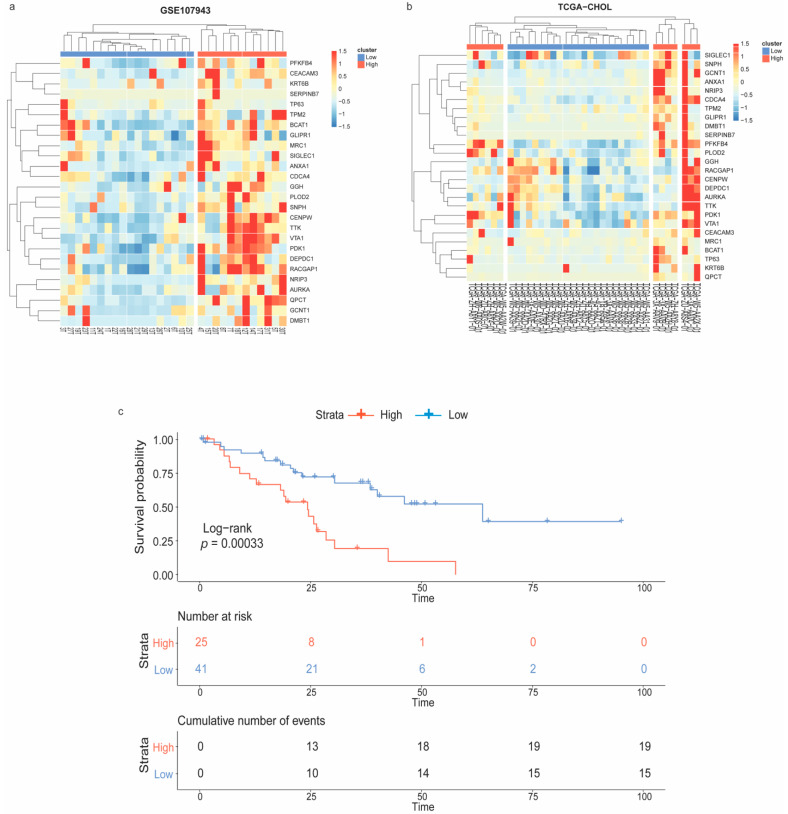
Patient stratification according to the immune-oncogenic gene signature and Kaplan–Meier survival analysis. Heatmap with unsupervised hierarchical clustering of the immune-oncogenic gene signature stratified CCA samples from (**a**) GSE107943 and (**b**) TCGA-CHOL into the low- vs. high-expression groups. (**c**) Kaplan–Meier survival curve revealed patients with the high-expression signature significantly associated with a poorer outcome.

**Figure 4 genes-13-00271-f004:**
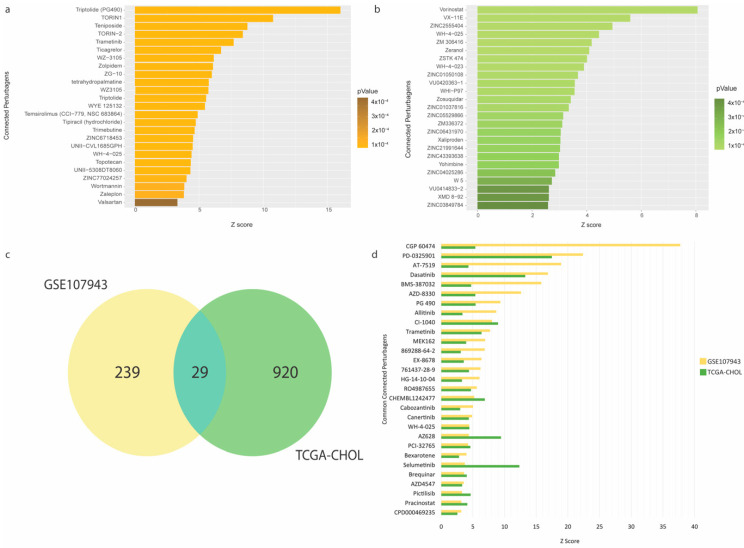
Pharmacogenomic connectivity analysis. (**a**) Top 25 connected perturbagens positively correlated to transcriptional changes between the low- vs. high-expression groups in the GSE107943 cohort. (**b**) Top 25 connected perturbagens positively correlated to transcriptional changes between the low- vs. high-expression groups in the TCGA-CHOL cohort. (**c**) Venn diagram depicting 29 commonly significant perturbagens between both cohorts. (**d**) Barplot showing 29 commonly significant perturbagens ranked by the connectivity z-score. Full results of pharmacogenomic connectivity mapping are provided in Appendix A.

**Figure 5 genes-13-00271-f005:**
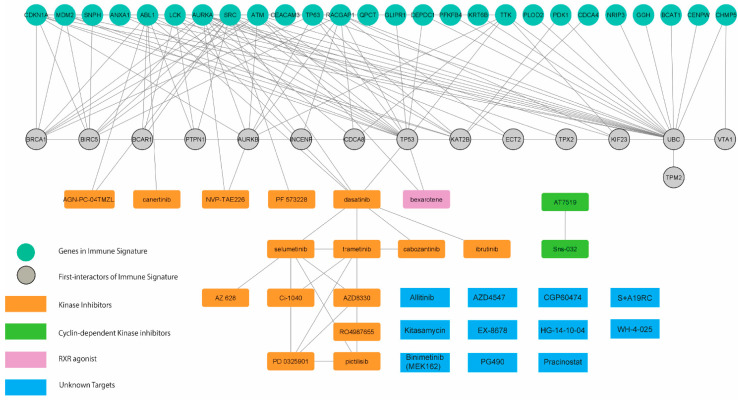
The integrative drug–gene network analysis. Interactions among 29 commonly connected perturbations and 26 mortality-associated immune-related genes were established depending on STITCH database using Cytoscape v.3.8 and stringApps v.1.7.0 plugin.

**Table 1 genes-13-00271-t001:** Candidate drugs predicted to have therapeutic efficacy in the CCA patient subpopulation with the high-expression of the immune-oncogenic gene signature in tumor tissues. NA, not applicable.

Perturbagen	Drug Class	Drug Target	Signature Gene	FDA-Approval/Clinical Trial Phase (Clinicaltrials.gov Identifier)
Bexarotene	RXR agonist	RXRA, RXRB, RXRG	*MDM2*	Approved for skin manifestations of cutaneous T-cell lymphoma
Cabozantinib	Tyrosine kinase inhibitor	KDR, MET, KIT, FLT3, TIE-2, RET, AXL	NA	Approved for hepatocellular carcinoma and advanced renal cell carcinoma Investigated as a monotherapy for cholangiocarcinoma after progression on first line and second line therapy (NCT01954745)
Dasatinib	Tyrosine kinase inhibitor	ABL1, FYN, LCK, SRC, KIT, YES1, EPHA2, LYN, PDGFRB, BCR, HCK, FGR, FRK, BLK, SRMS	*ABL1*,* FYN*,* LCK*,* SRC*,* FRK*	Approval for chronic myeloid leukemia with Philadelphia chromosome-positive Investigated for isocitrate dehydrogenase (IDH)-mutant advanced intrahepatic cholangiocarcinoma (NCT02428855)
Binimetinib (MEK162)	MEK inhibitor	MAP2K1, MAP2K2	NA	Approved in combination with encorafenib for unresectable/metastatic melanoma with BRAF V600E or V600K variants
Ibrutinib (PCI-32765)	BTK inhibitor	BTK	NA	Approved for B cell malignancies
Mirdametinib (PD-0325901)	MEK inhibitor	MAP2K1, MAP2K2	NA	Approved for neurofibromatosis type 1
Selumetinib	MEK inhibitor	MAP2K1, MAP2K2	NA	Approved for neurofibromatosis type 1 Investigated for unresectable cholangiocarcinoma with Ras pathway activation (NCT00553332)
Trametinib	MEK inhibitor	MAP2K1, MAP2K2	NA	Approved for unresectable/metastatic malignant melanoma with BRAF V600E or V600K variants Investigated in combination with hydroxycholoroquine in KRAS mutated refractory cholangiocarcinoma (NCT04566133)
Allitinib	Tyrosine kinase inhibitor	EGFR, ERBB2	NA	Phase II (NCT04671303)
AT-7519	CDK inhibitor	CDK2, CDK1, CDK9, CDK4, CDK5, CDK6, CDK14, CDK11B, CDK8, CDK7, CDK3, CDK16, CDK17, CDK18, CDK13, CDK10, CDK20, CDK15, CDK19, CDK12	NA	Phase I–phase II (NCT01183949, NCT02503709, NCT01652144, NCT01627054, NCT00390117)
AZD4547	FGFR inhibitor	FGFR1, FGFR2, FGFR3, FGFR4	NA	Phase I/II-phase II/III (NCT04439240, NCT02965378, NCT02824133, NCT01824901, NCT01791985, NCT01213160)
AZD-8330	MEK inhibitor	MAP2K1, MAP2K2	NA	Phase I (NCT00454090)
Canertinib	pan-EGFR inhibitor	EGFR, ERBB2, ERBB4	NA	Phase II (NCT00050830, NCT00051051, NCT00174356)
CI-1040	MEK inhibitor	MAP2K1, MAP2K2	NA	Phase II (NCT00033384, NCT00034827)
Triptolide (PG 490)	HSP70 inhibitor	NA	NA	Phase I–phase II (NCT03117920, NCT03129139)
Pictilisib	pan-PI3K inhibitor	PIK3CG, PIK3CD, PIK3R2, PIK3R1, PIK3CA, PIK3CB, PIK3R5, PIK3R3	NA	Phase I–phase II (NCT00975182, NCT00876122, NCT01740336, NCT02389842, NCT00876109, NCT00960960, NCT01493843)
Pracinostat	HDAC inhibitor	HDAC	NA	Phase I–phase III (NCT01912274, NCT03151408, NCT03848754, NCT01112384, NCT01075308, NCT00741234)
RO4987655	MEK inhibitor	MAP2K1, MAP2K2	NA	Phase I (NCT00817518)
SNS-032 (BMS-387032)	CDK inhibitor	CDK2, CDK7, CDK9	NA	Phase I (NCT00446342, NCT00292864)
NVP-TAE226 (761437-28-9)	Phenylmorpholines	FAK, InsR, IGF-1R, ALK, MET	*AURKA*	No entry in clinical trials yet
PF-573228 (869288-64-2)	Tyrosine kinase inhibitor	FAK, Pyk2, CDK1, CDK7, GSK-3β	*SRC*	No entry in clinical trials yet
AZ628	pan-Raf inhibitor	BRAF, RAF1	NA	No entry in clinical trials yet
CGP 60474	CDK inhibitor	CDK1	NA	No entry in clinical trials yet
CHEMBL1242477	NA	NA	NA	No entry in clinical trials yet
Kitasamycin (CPD000469235)	NA	NA	NA	No entry in clinical trials yet
EX-8678	NA	NA	NA	No entry in clinical trials yet
HG-14-10-04	NA	NA	NA	No entry in clinical trials yet
S+A19RC	NA	NA	NA	No entry in clinical trials yet
WH-4-025	Tyrosine kinase inhibitor	LCK, SRC, p38α, KDR	LCK, SRC	No entry in clinical trials yet

## Data Availability

All datasets and the R script that support the findings of this study are available in the Appendix A.

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
