# Peer review of "Molecularly Guided Drug Repurposing for Cholangiocarcinoma: An Integrative Bioinformatic Approach"

_genes, 2022, doi:10.3390/genes13020271_

Round 1

Reviewer 1 Report

I read the manuscript entitled as "Molecularly-guided drug repurposing for cholangiocarcinoma: An integrative bioinformatic approach" which has recently submitted to Genes with a great interest. 

This manuscript is well-written. This manuscript could provide those who treat patients with iCCA to useful information.  

Reviewer 2 Report

Using a transcriptomic-driven bioinformatic approach Venkemann et al. aim to find new therapeutic options to treat cholangiocarcinoma by repurposing drugs already approved by FDA for other diseases. This bioinformatic study is definitely interesting, novel and original, and the manuscript is clear and concise. 

Reviewer 3 Report

In this study the authors screened more than 3,000 immune related genes and identified 26 common genes having prognostic values in two cholangiocarcinoma (CCA) RNAseq deatasets. They also performed pharmacogenomics analysis and identified a panel of FDA approved drugs which may be useful to treat CCA in clinic.  The study is well designed and elegant and the findings are interesting and have the potential to be translated into clinical interventions.

Major critiques

  1. Figure 2a/b, the candidate prognostic genes shown in figure 2a/b do not match the number described in the text (e.g. TCGA-CHOL has 100+ prognostic genes, only a few on figure).
  2. The rationale to combine all 26 common prognostic gene into a 26-gene signature is not valid. This is because each gene may have its own expression pattern within the population and when combined together the signature may not function in sync. In fact, this can be exemplified by Figure 3, where the 26-gene signature can be roughly divided into two signatures.  One of these two is clearly a cell cycle signature ((AURKA, TTK, RACGAP1, CENPW, CDCA4) , which is a well-known poor prognostic signature in multiple cancer types.
  3. In this regard, the name “immune gene signature” used by the authors is inappropriate and misleading, because these cell cycle genes are not specific for immune cells and are likely expressed by the tumor cells. The authors also mentioned in their text that this signature is “oncogenic”, which is not normally associated with immune gene functions. The description, naming, and discussion of this result should be edited to address this issue.
  4. The authors should consider breakdown 26-gene signature into two parts and study whether any of these are related to tumor themselves or immune cells (tumor microenvironment), and repeat Figure 3c (the two sub-signatures may have different prognostic values).

Minor critiques

  1. Figure 2a/b, it will be helpful to mark common prognostic genes in 2a/b in a different color from non-commonly found genes.
  2. Figure 4 and Figure 5 may be put into supplemental results. The commonly changed pathways from these two independent studies can be combined to generate a new figure.
  3. Methodology description of L1000 related studies can be more detailed (e.g., reference 48 non-existent)

Round 2

Reviewer 3 Report

The authors have modified the manuscript to address my critiques.  The manuscript is improved.  I have no further questions.